# Deciphering Winter Sprouting Potential of *Erianthus procerus* Derived Sugarcane Hybrids under Subtropical Climates

**DOI:** 10.3390/plants13071023

**Published:** 2024-04-03

**Authors:** Mintu Ram Meena, K. Mohanraj, Ravinder Kumar, Raja Arun Kumar, Manohar Lal Chhabra, Neeraj Kulshreshtha, Gopalareddy Krishnappa, H. K. Mahadeva Swamy, A. Suganya, Perumal Govindaraj, Govind Hemaprabha

**Affiliations:** 1ICAR Sugarcane Breeding Institute, Regional Centre, Karnal 132011, India; ravinder.kumar6@icar.gov.in (R.K.);; 2ICAR Sugarcane Breeding Institute, Coimbatore 641007, India; r.arun@icar.gov.in (R.A.K.); gopalareddy.k@icar.gov.in (G.K.); mahadevaswamy.hk@icar.gov.in (H.K.M.S.); a.suganya@icar.gov.in (A.S.); p.govindaraj@icar.gov.in (P.G.); g.hemaprabha@icar.gov.in (G.H.); 3ICAR-Central Soil Salinity Research Institute, Karnal 132001, India; kulshreshthan@yahoo.com

**Keywords:** wild genetic resources, *Erianthus procerus*, winter sprouting potential, red rot resistance, subtropical climate

## Abstract

Winter sprouting potential and red rot resistance are two key parameters for successful sugarcane breeding in the subtropics. However, the cultivated sugarcane hybrids had a narrow genetic base; hence, the present study was planned to evaluate the *Erianthus procerus* genome introgressed Saccharum hybrids for their ratooning potential under subtropical climates and red rot tolerance under tropical and subtropical climates. A set of 15 *Erianthus procerus* derived hybrids confirmed through the *5S rDNA* marker, along with five check varieties, were evaluated for agro-morphological, quality, and physiological traits for two years (2018–2019 and 2019–2020) and winter sprouting potential for three years (2018–2019, 2019–2020, and 2020–2021). The experimental material was also tested against the most prevalent isolates of the red rot pathogen in tropical (*Cf671* and *Cf671* + *Cf9401*) and subtropical regions (*Cf08* and *Cf09*). The *E. procerus* hybrid GU 12—19 had the highest winter sprouting potential, with a winter sprouting index (WSI) of 10.6, followed by GU 12—22 with a WSI of 8.5. The other top-performing hybrids were as follows: GU 12—21 and GU 12—29 with a WSI of 7.2 and 6.9, respectively. A set of nine *E. procerus*-derived hybrids, i.e., GU04 (28) EO—2, GU12—19, GU12—21, GU12—22, GU12—23, GU12—26, GU12—27, GU12—30, and GU12—31, were resistant to the most prevalent isolates of red rot in both tropical and subtropical conditions. The association analysis revealed significant correlations between the various traits, particularly the fibre content, with a maximum number of associations, which indicates its multifaceted impact on sugarcane characteristics. Principal component analysis (PCA) summarised the data, explaining 57.6% of the total variation for the measured traits and genotypes, providing valuable insights into the performance and characteristics of the *Erianthus procerus* derived hybrids under subtropical climates. The anthocyanin content of *Erianthus procerus* hybrids was better than the check varieties, ranging from 0.123 to 0.179 (2018–2019) and 0.111 to 0.172 (2019–2020); anthocyanin plays a vital role in mitigating cold injury, acting as an antioxidant in cool weather conditions, particularly in sugarcane. Seven hybrids recorded a more than 22% fibre threshold, indicating their industrial potential. These hybrids could serve as potential donors for cold tolerance and a high ratooning ability, along with red rot resistance, under subtropical climates.

## 1. Introduction

Sugarcane, a perennial crop in the genus *Saccharum*, has served as a key feedstock for sugar production across the globe [1]. In light of climate change, it is becoming increasingly important to replace fossil fuels with sustainable bioenergy sources such as sugarcane [2,3]. In response to growing interest in biomass and ethanol, bioenergy traits have gained importance in recent sugarcane breeding programmes. The present commercial sugarcane hybrids are complex polyploidy aneuploids, primarily derived from the interspecific hybridisation between *Saccharum officinarum* and *Saccharum spontaneum.* The modern sugarcane cultivars contain 70–80% of the genome of *Saccharum officinarum*, which contributes to thick stalks and a high sucrose content [4]. Conversely, *Saccharum spontaneum* incorporates a high fibre content, high tillering, ratooning, and resistance to various biotic and abiotic stresses in the present-day cultivars [5]. However, the slow rate of genetic gain and narrow genetic base pose an immediate concern for sugarcane researchers, which necessitates the broadening of its genetic base using the allied genera. Therefore, attempts have been made to cross the *Saccharum* with allied genera to broaden the genetic base of future sugarcane varieties.

*Erianthus* is part of the ‘*Saccharum* complex’ as classified by Mukherjee in 1957 [6], along with four additional interbreeding genera, i.e., *Saccharum* L., *Miscanthus sect*. *Diandra Keng*, *Narenga* Bor., and *Sclerostachya* Hack. There is a notable emphasis on harnessing the potential of the *Erianthus* genus, a wild relative of *Saccharum*, to introgress the desired agronomic traits. Among the *Erianthus* genus, two notable species, namely, *Erianthus arundinaceus* and *Erianthus procerus*, have gained attention for their agronomically desired traits. These include high biomass, profuse tillering, robust ratooning potential, and resilience to both biotic and abiotic stresses. However, the challenges encountered in intergeneric crosses between *Saccharum* and *Erianthus* due to difficulty in the hybridisation process, cross-incompatibility, and the sterility of F_1_ hybrids impede further advancement of these efforts. Nevertheless, after subsequent attempts, the successful production of intergeneric hybrids from *Erianthus* has been documented in previous studies [7,8,9,10]. The lack of vegetative cane and large droopy silky panicles in *Erianthus procerus* (2n = 40) makes it different from *Erianthus arundinaceus* (2n = 30, 40, and 60) and *Erianthus kanashiroi Ohwi* (2n = 60) [11]. Efforts have been made at the ICAR-Sugarcane Breeding Institute, Coimbatore, India, to generate novel hybrids from *Erianthus procerus* and *Saccharum officinarum* [12]. Assessment of promising hybrids for their ability to exhibit high winter ratooning and frost tolerance is standard practice, particularly in regions that face low-temperature conditions. In India, the mega-sugarcane cultivating areas belong to the northwest zone of the country, characterised by a subtropical climate, where the sugarcane crop is planted during the autumn season, subjected to chilling (<10 °C) and sub-zero temperatures in December-January, resulting in very slow growth or negligible cane growth and sprouting. When the sugarcane crop is ratooned during the peak winter season, many popular varieties are not capable of providing better ratooning, resulting in a 10–20% reduction in ratoon yields as compared to the plant crop [13]. Also, multi-ratooning is not gaining popularity due to the harvesting of preceding crops coinciding with the winter period, which affects ratoon sprouting. Hence, there is utmost significance in the improvement of the winter sprouting potential to increase the cane yield and sugar recovery in subtropical regions. Additionally, autumn planting with multi-ratooning is an important factor in improving the sugarcane yield and sugar recovery in the subtropical climate of India [14]. The integration of the wild genome in the resultant hybrids has shown improved resilience to water scarcity, increased biomass yield, resistance to pests and diseases, and adaptability to varied environmental stresses [12]. In subtropical climates, emphasis is given to the identification of sugarcane varieties with higher yields, increased sugar content, and resistance to diseases. Additionally, priority is set to those varieties that exhibit a higher winter ratooning ability. Therefore, germplasms with better winter sprouting potential are viewed as a way forward to address the problem of winter ratooning in subtropical sugarcane breeding programmes.

The present study was planned to evaluate the *Erianthus procerus* genome-introgressed *Saccharum* hybrids for their ratooning potential under subtropical climates and red rot tolerance under tropical and subtropical climates. Also, to study the variation in the *Erianthus procerus* genome-derived hybrids for agro-morphological, quality, and physiological parameters. These intergeneric hybrids could serve as valuable sources for the further introgression of high ratooning potential as well as biotic and abiotic stress tolerance into the cultivated sugarcane.

## 2. Results

### 2.1. Weather Statistics of the Experimental Site

The maximum and minimum temperatures, evaporation rates, relative humidity, and rainfall patterns of subtropical India from January 2018 to December 2020 are presented in Figure 1. In all three years, the pattern of maximum and minimum temperatures was generally found to be similar. The monthly maximum and minimum temperature ranged between 25–45 °C and 10–30 °C, respectively. However, during peak winter (the second fortnight of December to the first fortnight of January), the minimum temperatures ranged from 4 to 8 °C. The rainfall pattern suggests that the July–August months were the wettest months in all the years; however, the distribution varied among the years. Active rainfall months were June–September in the year 2018 and June–August in the year 2020; however, in the year 2019, only July–August were the active rainfall months. In subtropical conditions, the sunshine hours also varied to a great extent, ranging from as low as 3.4 h/day (December 2019) to as high as 10.8 h/day (May 2018), which exactly corresponds to the temperature patterns and inverse relationship with the relative humidity pattern.

### 2.2. Significance of Variation Sources

The pooled ANOVA was carried out after conducting the homogeneity of residual variance test to divide the total variation in the traits into its components, and the results are presented in Table 1. The ANOVA revealed that the first main effect, i.e., the genotype effect, was highly significant (*p* < 0.01) for all the agro-morphologies (60-day tillers, 90-day tillers, 120-day tillers, plant height, cane diameter, and cane yield) except germination and the number of millable canes, which were significant at *p* < 0.05. Similarly, quality trait (juice weight, juice extraction, Brix value, pol %, purity, single cane weight, and fibre content) parameters were highly significant at *p* < 0.01 (Table 1). However, out of the five physiological parameters, only two parameters, i.e., the nitrogen balance index and chlorophyll content, exhibited a highly significant (*p* < 0.01) genotypic effect, whereas the remaining three parameters, i.e., the flavonoid index, anthocyanin index, and leaf area, did not exhibit significant differences among the studied traits. The other main effect, i.e., the year effect, was highly significant (*p* < 0.01) for germination, 90-day tillers and 120-day tillers, and significant (*p* < 0.05) for the pol% and fibre content, and the remaining 14 traits exhibited no significant differences among the studied genotypes. The interaction component (genotype × year) effect was highly significant (*p* < 0.05) only for the 90-day tillers, 120-day tillers, and fibre content, whereas the remaining 17 studied traits did not exhibit significant differences among the genotypes.

### 2.3. Mean Performance

#### 2.3.1. Agro-Morphological and Quality Traits

The mean performance of agro-morphological, quality, and physiological parameters is illustrated as box plots in Figure 2, Figure 3 and Figure 4 and presented in the Appendix A. Although all the studied traits were influenced by the environment, the magnitude of the environmental effect was higher for the agro-morphological traits compared to the quality and physiological traits. The magnitude of variation for the number of tillers at 60 days was more than that of 90 days and 120 days. The mean germination was also highly influenced by the environmental effect. Generally, the crop year 2018–2019 was more favourable for all the studied traits, except for the germination and flavonoid index, than the crop year 2019–2020, as the trait’s performance measured by the percent superiority ranged from 1.8 to 54.7. The extreme genotypes were observed for a few traits, including juice extraction, purity, fibre content, cane diameter, and cane yield, as some of the genotypes were located outside the error bars (Figure 2, Figure 3 and Figure 4).

In the year 2018–2019, the plant height varied from 136.67 cm to 263.48 cm with a mean of 136.67 cm, and many procerus-derived hybrids were taller than the best modern cultivar, Co 0238 (Appendix A). The cane diameter ranged from 1.50 cm to 2.97 cm with a mean of 2.01 cm; unlike the cane weight, the cane diameter reduced in many of the procerus-derived hybrids. The tillers at 60 days after planting and 90 days after planting were high in GU 12—30, with a mean of 131.95 and 138 ‘000/ha, respectively. In the year 2019–2020, the plant height varied between 120.12 and 263.33 cm with a mean of 200.73 cm, and the cane diameter varied from 1.08 cm to 2.84 cm with a mean of 1.92 cm. The tillers at 60 days after planting and 90 days after planting varied between 50.37 and 200, and 54.81 and 104.07, with mean tiller numbers of 122.63 and 77.29 ‘000/ha, respectively. Out of these, five entries, namely, GU12—29, GU12—28, GU12—30, GU12—34, and GU12—38, had superior performance compared to the best standard, Co 0238 (150 ‘000/ha), and the other five entries, namely, GU12—21, GU12—23, GU12—22, GU12—26, and GU12—27, were recorded at par value with the check Co 0238. The cane yield in the years 2018–2019 and 2019–2020 ranged from 35.15 to 99.16 t/ha and 38.12 to 92.85 t/ha, respectively, with the mean yields of 69.22 and 64.32 t/ha. The juice pol percent varied from 9.14 to 18.21 percent in 2018–2019 and from 11.36 to 16.83 percent in 2019–2020 with a mean of 14.64 and 13.96 percent, respectively. Similarly, the mean leaf area during the year 2018–2019 was 243.06 cm^2^, it was 237.19 cm^2^ during 2019–2020, and Co 0238 was best among the standards with a 241 cm^2^ leaf area.

#### 2.3.2. Physiological Traits

##### Flavonoid and Anthicyanin Indices

The flavonoid index ranged from 0.89 to 1.38 (2018–2019) and 0.93 to 2.16 (2019–2020), whereas the mean flavonoid index of 1.14 and 1.23 was recorded, respectively, during the years 2018–2019 and 2019–2020 (Figure 4 and Appendix A). Among the studied clones, CoJ 64 and GU 12—30 recorded a higher flavonoid index of more than 1.14 in the year 2018–2019, while GU 12—21 and CoJ 64 recorded a better flavonoid index (>1.23) in the year 2019–2020. The GU12—19, GU12—33, GU12—19, and GU12—23 recorded a lower flavonoid index in the years 2018–2019 and 2019–2020, respectively. Similarly, the anthocyanin index ranged from 0.123 to 0.179 (2018–2019) and 0.111 to 0.172 (2019–2020), whereas the mean anthocyanin index of 0.147 and 0.144 was recorded, respectively, during the years 2018–2019 and 2019–2020 (Figure 4 and Appendix A). Among the studied clones, Co 0238 and GU 12—22 recorded a higher anthocyanin index of more than 0.147 in the year 2018–2019, while GU12—30 and GU12—38 recorded a better anthocyanin index (>0.144) in the year 2019–2020. The genotypes CoJ 64, GU 04(28), GU 12—34, and Co 0238 recorded a lower flavonoid index in the years 2018–2019 and 2019–2020, respectively.

##### Exploring Fibre Content in Procerus-Derived Hybrids

The fibre content for the industrial utilisation of the procerus hybrids was assessed and is presented in Figure 5. The mean fibre content of the trial was recorded at 19.66% and 18.43% in the years 2018–2019 and 2019–2020, respectively. Seven *Erianthus procerus*-derived hybrids, viz., GU12—30, GU12—28, GU12—38, GU12—33, GU12—19, GU12—31, and GU12—29, had a >22.0% fibre content, whereas three hybrids had a 20–22% fibre content and five hybrids had a 15–20% fibre content. A similar trend was also observed in the year 2019–2020 (Figure 5).

### 2.4. Correlation Studies

The correlation coefficients of the agro-morphological, quality, and physiological traits are given in Table 2. The cane yield had a significant positive association with the number of tillers at 60 days (0.57) and 90 days (0.39), the plant height (0.51), the single cane weight (0.84), the purity (0.35), and the fibre content (0.35); however, the juice weight had a significant negative association. The important quality trait, i.e., pol%, had a significant positive correlation with the cane diameter, juice weight, juice extraction, Brix value, purity, and single cane weight, and a significant negative association with the fibre content, number of millable canes, and nitrogen balance index. The fibre content had a significant association with the maximum number of 14 traits, including 7 traits with a positive association (number of tillers at 60, 90, and 120 days, plant height, number of millable canes, cane yield, and nitrogen balance index) and 7 traits with a negative association (germination, cane diameter, juice weight, juice extraction, Brix value, pol%, and purity). The other important economic trait, i.e., the juice weight, had a significant positive association with the cane diameter, juice extraction, Brix value, pol%, purity, and single cane weight. Similarly, juice extraction had a significant positive association with the cane diameter and juice weight, Brix value, pol%, purity, and single cane weight, and a significant negative association with the fibre content, number of millable canes, nitrogen balance index, and chlorophyll concentration.

Physiological traits had fewer significant associations with agro-morphological and quality traits. Out of the studied four physiological traits, the nitrogen balance index had significant associations with a maximum of eight traits, including three positive (plant height, fibre content, and nitrogen balance index) and five negative (juice extraction, Brix value, pol%, flavonoid index, and anthocyanin index) associations. The chlorophyll concentration had a significant positive association (0.84) with the nitrogen balance index and a significant negative association with juice extraction (−0.37) and purity (−0.36). The anthocyanin index had a significant negative association with the nitrogen balance index and the chlorophyll concentration, and a significant positive association with the flavonoid index. The physiological trait leaf area did not have a significant association with any of the studied traits. Generally, there was a strong and significant positive association among the sugarcane juice quality parameters, including the juice weight, juice extraction, Brix value, pol%, and purity.

### 2.5. Principal Component Analysis (PCA)—Biplot Analysis

The inter-relationship among various traits and genotypes was assessed in the form of biplots in each environment, as shown in Figure 6 and Figure 7. The PCA was performed on 20 traits and measured their contribution to the variability. In the year 2018–2019, five PCs were significant and accounted for 85.27% of the total variation. The first PC captured around 36.6% of the total variation, and that was attributed to the single cane weight, pol%, purity, juice extraction, juice weight, germination, and anthocyanin index for the largest positive loadings. The second PC captured 21.04% of the total variation, majorly contributed from the cane yield, tillers numbers, plant height, fibre content, and numbers of millable canes, with positive loadings. The biplot based on PC1 and PC2 placed the high-yielding genotypes in the first two quarters and also revealed the significant positive correlation of the cane yield with the plant height, single cane weight, and tiller numbers. In the year 2019–2020, seven PCs were significant and represented 89.08% of the total variability. The first PC captured around 28.01% of the variability and was attributed to the positive loadings of the cane diameter, juice weight, germination, and single cane weight, whereas the second component captured 19.5% of the total variability, represented by the cane yield, tiller numbers, plant height, and fibre content. Similar to 2018–2019, the first two-quarters of the biplot based on PC1 and PC2 were occupied by the high-yielding genotypes.

### 2.6. Winter Sprouting and Red Rot

#### 2.6.1. Winter Sprouting Potential in *Erianthus* Procerus Vis-a-Vis Standard Checks

The winter sprouting ability, measured as the winter sprouting index (WSI) of *Erinathus procerus* derived hybrids and check varieties, is presented in Figure 8A and Table 3. The WSI of all the check varieties was lower than the *Erinathus procerus* derived hybrids used in the study. The WSI of the commercial check variety ranged from 1.4 to 2.8. However, the *Erinathus procerus* derived hybrids ranged from 3.4 to 10.6. Among the check varieties, Co 0238 was the best check, followed by CoJ 64. However, the best check (Co 0238) was inferior to the *Erinathus procerus* derived hybrids. The percent superiority of *Erinathus procerus* derived hybrids over the best commercial check (Co 0238) is given in Figure 8B. GU 12—19 was the best genotype for winter sprouting ability, with a WSI of 10.6 and a high percent superiority (279%), followed by GU 12—22 (WSI: 8.5 and a 204% superiority over the best check), GU 12—21 (WSI: 7.2 and a 157% superiority over the best check), and GU 12—29 (WSI: 6.9 and a 146% superiority over the best check). Out of the 15 hybrids, nine *Erinathus procerus* derived hybrids had a better winter sprouting ability over the best check, Co 0238, and the experimental mean (WSI: 3.8); however, the remaining six hybrids had better performance over all the checks and were also on par with the performance of the experimental mean value of WSI, as indicated in Figure 8A.

#### 2.6.2. Resistance of *Erianthus procerus*-Derived Hybrids against the Red Rot

Red rot caused by *Colletotrichum falcatum* is an important stalk disease of sugarcane, causing substantial yield losses in India and other countries. Host-plant resistance plays a vital role in managing the disease. Traditionally, *Saccharum spontaneum* has been the main source of red rot and other abiotic stress tolerances. The development of new variations in *Colletotrichum falcatum* is also a major concern and poses challenges in the cultivation of newer varieties. The exploitation of new and diverse genetic resources is essential to enhance disease resistance. Of late, considerable attention has been given to the incorporation of the wild genus *Erianthus*, which has been identified as a valuable source of many traits, including red rot resistance. In this study, 15 hybrids were subjected to the screening process against subtropical and tropical isolates of *Colletotrichum falcatum*. The results of the screening revealed varying levels of resistance among the tested clones. A high level of resistance (R) to red rot against virulent isolates of subtropical *Cf08* and *Cf09* was observed, indicating their ability to withstand infection and exhibit minimal disease symptoms. The clones were classified as moderately resistant (MR), indicating a moderate level of resistance to the pathogen. On the other hand, the clones were categorised as moderately susceptible (MS), indicating a higher susceptibility to red rot compared to the resistant and moderately resistant clones. The last category includes clones that are classified as susceptible (S), indicating their vulnerability to infection and the development of severe disease symptoms (Table 4). The red rot rating of the F_1_ hybrid and BC hybrids against the mixed inoculum of *Cf671* and *Cf94012* (*Colletotricum falcatum*) is given in Table 4. The intergeneric hybrids have shown resistant reaction and the back-cross hybrids, viz., GU 12—21, GU 12—23, GU 12—29, and GU 12—31, were also resistant, and other BC hybrids were moderately resistant.

## 3. Discussion

The genetic resources of the *Erianthus* genus consist of seven species; among them, two species, i.e., *Erinathus arundinaceus* and *Erinathus procerus*, are the most promising. The genus *Erianthus* gained significant attention from sugarcane breeders worldwide as a source for high tillering, high ratooning ability, high biomass, and tolerance against potential biotic and abiotic stresses [9,12,15]. The *Erianthus* genus is highly adapted to varied environments and is extensively distributed in both tropical and subtropical regions [16]. However, the introgression of *Erianthus* had limitations due to low compatibility, high genetic distance, and difficulty in identifying the true hybrids from the selfed ones. The work on the utilisation of *Erianthus arundinaceus* and its introgression into sugarcane for higher biomass has been successful [16,17,18,19]. Similarly, the utilisation of *Erinathus procerus* in sugarcane breeding was very limited, and the work on the intergeneric hybridisation between *Saccharum* and *Erinathus procerus* was performed at the ICAR-Sugarcane Breeding Institute, Coimbatore [20]. The developed intergeneric hybrids were confirmed using *5S rDNA* markers, which can differentiate between *Saccharum* and *Erianthus* [7,21,22]. The confirmed true intergeneric hybrids were again back-crossed to *Saccharum* spp. to recover more of the *Saccharum* genome. A total of 15 such clones derived from the programme of intergeneric hybridisation and back-crossing were evaluated for their agronomic performance, winter ratooning ability, and reaction against red rot resistance.

### 3.1. Variability and Mean Performance for Various Economic Traits

The genotypic effect for all the agro-morphological and quality traits was significant, indicating the importance of *Erinathus procerus*-derived breeding lines in commercial breeding programmes to improve the economically important traits, such as the tiller number, plant height, cane diameter, and number of millable canes, which ultimately decide the cane yield. The year effect is significant only for six traits, including the fibre content and pol%, which indicated the inconsistent nature of these traits. The interaction was significant for tillers at 90 and 120 days and the fibre content, indicating that the selection for these traits should be done cautiously with multi-environment testing.

A positive association was observed between the cane yield and the number of tillers at 60 and 90 days, the plant height, and the single cane weight; previously, similar results were reported in commercial canes [23]. The procerus hybrids were tall-growing, but they were thinner than the commercial canes; hence, in our study, the diameter did not influence the cane yield. However, the juice weight had a significant negative association with the cane yield, indicating more fibre content in the procerus-derived clones. It was concluded that the extraction percentage was also negatively associated with the fibre content and quality traits, including the pol% and Brix value. The genetic variability of the clones for the fibre percentage and its related traits was observed in the *Erinathus procerus* hybrids [24].

PCA is a convenient statistical technique that found application in the reduction of the original variables into a smaller number of underlying variables to disclose the inter-relationships among various traits and genotypes in each environment, as shown in Figure 6 and Figure 7. Through the biplot analysis, four kinds of information can be inferred, i.e., the relationship among the genotypes, the trait variability, the correlation among traits, and the relative value of the accession to the character. The longer the line in the PCA, the higher the variance. The cosine of the angle between the lines denotes the correlation between the variables. The closer the angle is to 90 or 270°, the smaller the correlation, and an angle of 0 or 180° suggests a correlation of 1 or −1, respectively [25,26].

### 3.2. Winter Sprouting Potential

The winter sprouting ability, measured as the winter sprouting index (WSI), is very important for sugarcane varieties grown in subtropical regions. Unlike tropical climates, subtropical regions face many climatic vagaries, viz., high temperature and water-deficit stress during summer and chilling injury–cold stress during winter [27]. The crop is also affected by water-logging during the rainy season. Sugar mills in the subtropical regions start cane crushing in mid-November; therefore, the harvesting of the crop starts in November and continues until January for the early crushing periods; this period coincides with peak winter. The harvesting of early-maturing sugarcane varieties in the region starts during the peak winter period when temperatures come down near 2–7 °C. The harvesting of the cane during this period reduces their sprouting due to the chilling weather conditions prevailing over a two-month period. The harsh winter reduces the crop ratooning ability due to lack of sprouting in low temperatures (4–10 °C) and is one of the main reasons for the low productivity in the ratoons of the subtropics [28]; moreover, farmers are not able to harvest the full potential of the ratoon crop. Any sugarcane genotypes that sprout during such chilling temperatures, harbour cold tolerant genes, and such genotypes are considered as highly winter tolerant with a good ratooning potential. The introgression of cold-tolerant genes conferring a high ratooning ability from allied genera is a promising alternative to the breeding of suitable cultivars with a better winter ratooning ability. In the present study, *Erianthus procerus*-derived hybrids were evaluated for their winter sprouting potential in subtropical climates. The nine procerus hybrids demonstrated a notably superior winter sprouting ability compared to the top-performing check Co 0238. The genotypes GU 12—19, GU 12—22, and GU 12—21 reported a superior WSI and will serve as an alternate source of donor parent to introgress the winter sprouting ability in subtropical varieties and enhance the better ratooning ability of the commercial clones under cold-stress conditions. The identified genotypes with a higher winter ratooning potential could serve as a better option for the sugarcane breeder to utilise these clones in breeding programmes to improve winter tolerance in the cultivated sugarcane varieties. The ultimate aim of the breeder is to improve the productivity of the sugarcane varieties under challenging environmental conditions. The earlier study with 632 sugarcane germplasms revealed a significant and positive correlation of the winter sprouting index (WSI) with the numbers of tillers and millable canes in the ratoon crop [14,29,30,31]. The highest WSI of 6.5 was also reported, and in this study, five clones had a WSI of more than 6.5, indicating the superior winter ratooning ability of the *Erinathus procerus*-introgressed hybrids. This study also further confirmed the utilisation of the WSI to assess the winter ratooning ability of sugarcane clones and identified the noteworthy procerus hybrids for better ratooning potential [27,32,33]. Our study further strengthened sugarcane breeding toward the identification of the desired high winter sprouting donors, which will be utilised in the future for the improvement of the high winter ratoonability under challenging environments.

### 3.3. Red Rot Resistance

Red rot in sugarcane is the major problem of sugarcane cultivation in the subtropics and is responsible for the loss of several elite varieties from cultivation [34]. In the present study, the *Erinathus procerus* derived hybrids were also screened for their reaction against tropical and subtropical isolates of the red rot pathogen. The hybrid GU04 (28) EO—2 and the back-crossed hybrids GU12—19, GU12—21, GU12—22, GU12—23, GU12—26, GU12—27, GU12—30, and GU12—31 were R to MR for the tested isolates, indicating their broad-spectrum resistance against red rot. Three clones, viz., GU 12—19, GU 12—21, and GU 12—22, showed both an excellent winter ratooning ability and broad-spectrum red rot resistance, indicating their potential to improve both traits. Further crossing of GU 12—21 with commercial clones revealed that more than 60% of the progenies were resistant to red rot, suggesting the valuable potential of the clones in imparting red rot resistance. Similar findings were also reported in *Erinathus arundinaceus* introgressed clones with cold tolerance and red rot resistance [31,35]. The *Erinathus procerus* derived hybrids proved to be the wealthiest pre-bred lines, which can be utilised in the sugarcane improvement programme. Their high biomass nature, high tillering ability, winter sprouting ability, high fibre content, and red rot resistance give new hope to the further enhancement of the winter ratooning ability and red rot resistance in commercial sugarcane varieties with a broadened genetic base.

### 3.4. Fibre and Flavonoids in Erinathus procerus-Derived Hybrids

The fibre in sugarcane is important in the modern world as a source of energy to utilise in cogeneration, the production of second-generation bioethanol, biogas, and pyrolysis to produce bio-oil and biochar [36,37,38]. In our study, seven *Erinathus procerus* hybrids recorded more than 22% fibre, indicating that their proportion of fibre content could be processed to produce fibre suitable for weaving into textiles, such as bags, clothing, and home textiles, making them more suitable for industrial applications and bioenergy production [39]. Cane with a high fibre content can be utilised in the production of biodegradable materials, such as bioplastics and packaging, offering sustainable alternatives to traditional plastic products.

In the study clones, CoJ 64 and GU12—30 exhibited a higher flavonoid index, exceeding 1.14 in the year 2018–2019, while GU 12—21 showed superior flavonoids (>1.23) in the year 2019–2020. The significance of flavonoids in sugarcane has been previously documented by Colombo et al. [40]. Similarly, hybrids GU12—22 and Co 0238 exhibited a higher anthocyanin index, exceeding 0.147 in the year 2018–2019, while GU12—30 and GU12—38 showed a better anthocyanin index (>0.144) in the year 2019–2020. The role of anthocyanin accumulation in cool weather is crucial, acting as antioxidants against cold injury. Anthocyanin accumulation is positively correlated with antioxidant capacity, protecting against cold stress [20,41]. Anthocyanin also plays an inhibitory role in the production of free radicals and reducing the levels of reactive oxygen species (ROS) under low-temperature stress [42].

## 4. Materials and Methods

### 4.1. Experimental Site

The experiment was conducted at the ICAR-Sugarcane Breeding Institute, Regional Centre, Karnal (Haryana) India, which is located at 29.1–29.5° N and 76.3–77.1° E in a subtropical climate, with an elevation of 243 m above the mean sea level and an average annual rainfall of 744 mm. In summer, the maximum temperature ranges from 34 to 45 °C, while in winter, the minimum temperature ranges from 5 to 8 °C. The soil in this area varies from clay–loamy to loam, with a pH range of 8.0–8.5.

### 4.2. Plant Material and Experimental Design

A total of 15 clones of *Erinathus procerus*-derived hybrids were evaluated along with five standard sugarcane varieties, i.e., Co 0238, CoJ 64, CoS 8436, CoS 767, and Co 06027, for their cane yield, physiological traits, and winter sprouting potential. Also, the experimental material was evaluated for both tropical and subtropical isolates of red rot resistance. A list of the *Erinathus procerus*-derived hybrids along with their parentages are provided in Table 5. The date of planting for the plant crop was in the first week of March during the 2018–2019 crop season and harvested during March-April during 2019–2020. Similarly, the ratoon crop was raised in 2019–2020 during the spring season (Feb-March). The experiment was carried out over a span of two years (the 2018–2019 and 2019–2020 spring seasons) and three seasons for the WSI using a randomised block design with three replications. Plots utilised had dimensions of 2 rows × 3 m and were spaced 0.9 m apart. Standard recommended practices were followed for crop production. Three lifesaving irrigations were applied at the time of germination, tillering, and the grand growth stage of the crop; otherwise, the crop was managed with natural rainfall throughout the year. The observations were collected from each plot, specifically for the number of tillers and the numbers of millable canes, the Brix value, the single cane weight, the plant height, and juice quality parameters [15,35,43,44,45,46] and physiological parameters like the chlorophyll content, nitrogen balance index (NBI), and flavonoid and anthocyanin index, were recorded from the 10-month-old crop. Five randomly selected canes from each plot were tagged and assessed for the estimation of the fibre percentage; 250 g subsamples of shredded canes were crushed in a rapipol machine and subsequently oven-dried. The fresh and dry weights of these samples were recorded. The fibre percentage was determined using the rapipol extraction method, as outlined by Thangavelu and Rao [44].Fibre percent= (A−B)/C×100
where A represents the dry weight of bagasse (residue) plus the bag after the drying process (measured in grams), B represents the dry weight of the bag alone (measured in grams), and C represents the fresh weight of the cane (measured in grams).

To study the winter sprouting capability, the stalks of the plant crops were cut at ground level in three replications during the last week of December 2018, aligning with the peak winter period. This specific block of three replications was designated for observing the winter-initiated ratoon crop.

The winter sprouting index (WSI) was worked out following the formula suggested by Ram et al. [14], as follows:$$\mathrm{Winter}\;\mathrm{Sprouting}\;\mathrm{Index}\;=\;\frac{\left(\mathrm{\% of alive and}\;\mathrm{sprouted clumps per plot}-1\right)\times \mathrm{No of sprouted clump}-1)}{100}$$

Based on the WSI, the sugarcane genotypes were classified into four categories, as shown below.

WSI Sprouting Category

3.00 and above: Excellent winter sprouting;

2.01 to 2.99: Good winter sprouting;

0.10 to 2.00: Poor winter sprouting;

<0.10: Low temperature-sensitive (LTS) clones.

To compute the fresh biomass yield (measured in tonnes per hectare, t/ha), the formula used was fresh biomass yield = NMC/ha × SCWT (kg) with the green cane top.

### 4.3. Red Rot Evaluation

To assess their resistance to red rot, a comprehensive screening process was conducted on all *Erianthus* procerus-derived hybrids. This study aimed to evaluate their resistance against the predominant and highly virulent isolates of red rot, *Colletotrichum falcatum*, specifically *Cf08* and *Cf09* in the subtropics and *cf671* and *cf671* + *cf94012* in the tropics [35,47]. The screening was performed under field conditions. To initiate the screening, a mixed inoculum containing both isolates of *Colletotrichum falcatum* was prepared. This inoculum was then applied to the sugarcane plants through plug and nodal methods. The screening was conducted when the crops were approximately seven months old, with inoculation occurring in September.

### 4.4. Measurement of Physiological Parameters

The leaf analyser Dualex Scientific+ (Force-A, Orsay, Paris, France) was used to estimate the polyphenol and chlorophyll content of the leaves based on the absorbance and transmittance of specific wavelengths of visible and near-infrared light [44]. The instrument does not directly measure these chemical concentrations, but measures the reflectance and absorbance indices that have been demonstrated to relate to the leaf-level concentrations. The nitrogen balance index and flavonoid, chlorophyll, and anthocyanin contents were observed from the top visual dewlap (TVD) of three different plants.
$$Chlorophyll\;concentration\;(\mu g/{cm}^{2})=\frac{NIR-Red}{Red}$$
$$Flavonoid\;Index=log\frac{NIR\;fluo\;excited\;Red}{NIR\;fluo\;excited\;UA-A}$$
$$Anthocyanin\;Index=log\frac{NIR\;fluo\;excited\;Red}{NIR\;fluo\;excited\;green}$$
$$Nitrogen\;Balance\;Index=\frac{Chlorophyll}{Flavonoids}$$

### 4.5. Confirmation of Intergeneric Hybrids

*Erianthus*-specific markers were developed from *5S rRNA* spacer regions using a sequence-tagged PCR. An *Erianthus procerus* clone along with randomly selected intergenic hybrids and the *Saccharum* spp. hybrid Co 86032 (commercial variety) were screened with primers that amplified the *5S rDNA* regions in the genomes to identify the *Erianthus*-specific fragments (Figure 9). The PCR amplification conditions for the *5S rDNA* of D’Hont et al. [7] were followed. Amplified products were resolved on 2% agarose gels stained with ethidium bromide and documented using a gel documentation system (Syngene, Bangalore, India).

### 4.6. Statistical Analysis

The collected data underwent analysis of variance (ANOVA) tests for statistical evaluation [48]. The statistical package SAS 9.3 software (SAS Institute Inc., Cary, NC, USA) was utilised to calculate the means, standard deviations, and coefficients of variance for various traits [49].

## 5. Conclusions

The longevity of the sugarcane cultivar in the subtropical regions depends on the winter sprouting potential and broad-spectrum red rot resistance, along with the yield and quality. This study identified two potential clones (GU12—19 and GU12—21) for their high WSI and red rot resistance. Further, GU12—19 contains >22.0% fibre, making it more suitable for industrial applications and bioenergy production. *Erianthus procerus*-derived hybrids exhibited superior performance for physiological parameters, including anthocyanin, which plays a vital role in mitigating cold injury. The cultivated sugarcane hybrids had a narrow genetic base for tolerance to cold injury and winter sprouting ability in red rot resistance genetic backgrounds. Hence, the *Erianthus procerus*-derived hybrids proved to be the wealthiest pre-bred lines, which can serve as a potential source to be utilised in future breeding programmes to develop sugarcane cultivars with red rot resistance and a high winter ratooning potential in subtropical conditions.

## Figures and Tables

**Figure 1 plants-13-01023-f001:**
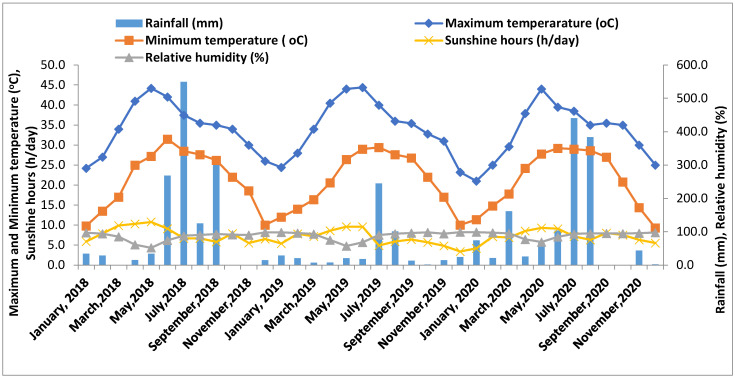
The weather parameters during January 2018 to December 2020.

**Figure 2 plants-13-01023-f002:**
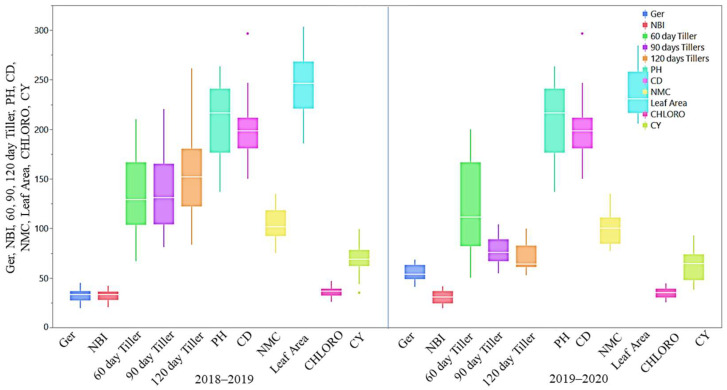
Box plots for agro-morphological, quality, and physiological traits of *Erianthus Procerus* derived hybrids and check varieties. Ger: germination (%); NBI: nitrogen balance index; 60-day tillers (‘000/ha); 90-day tillers (‘000/ha); 120-day tillers (‘000/ha); PH: plant height (cm); CD: cane diameter (cm); NMCs: number of millable canes (‘000/ha); leaf area (cm^2^), CHLORO: chlorophyll concentration (µg/cm^2^); CY: cane yield (t/ha).

**Figure 3 plants-13-01023-f003:**
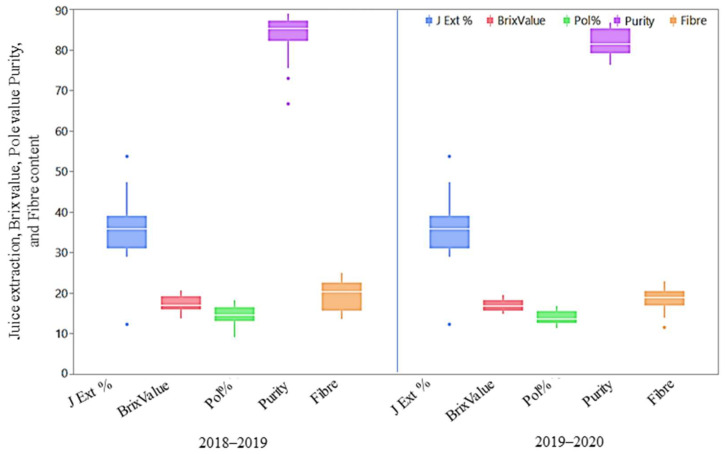
Box plots for juice extraction (%), Brix value (%), pol percent (%), purity (%), and fibre content (%) of *Erianthus procerus* derived hybrids and check varieties.

**Figure 4 plants-13-01023-f004:**
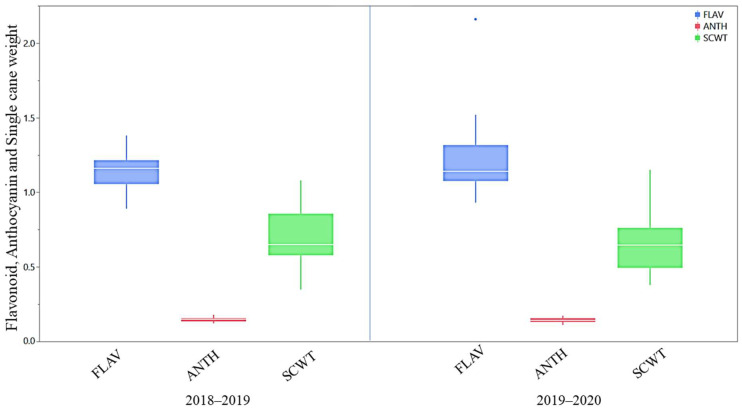
Box plots for flavonoid, anthocyanin, and single cane weight of *Erianthus procerus* hybrids and check varieties. FLAV: flavonoid index, ANTH: anthocyanin index, SCWT: single cane weight (kg).

**Figure 5 plants-13-01023-f005:**
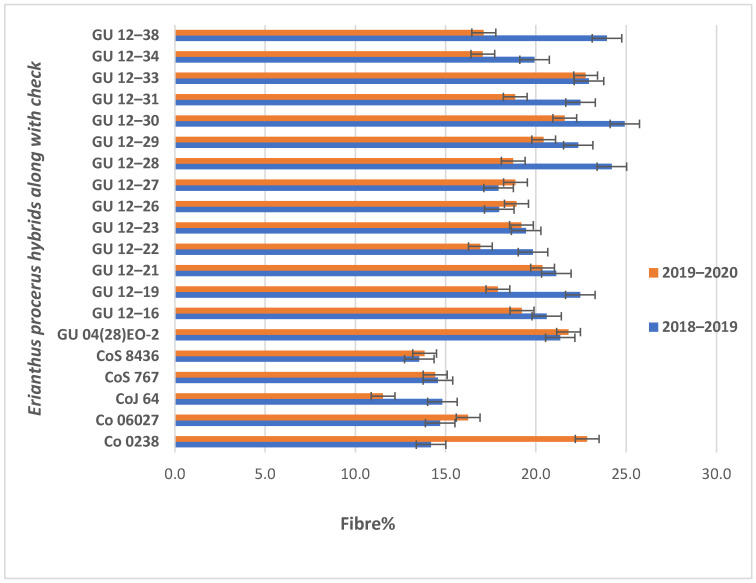
Fibre percent in *Erianthus procerus*-derived hybrids and check varieties.

**Figure 6 plants-13-01023-f006:**
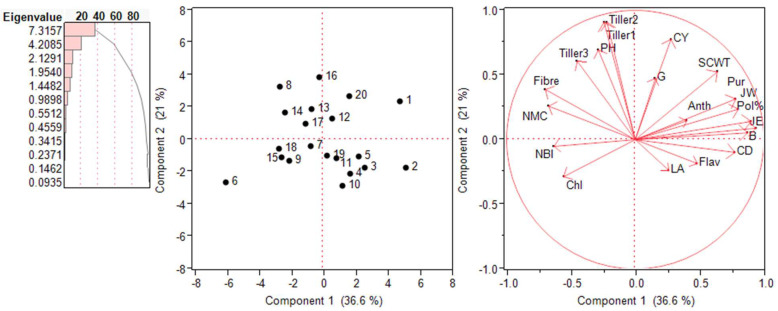
Scree plot, score plot, and loading plots of PCA1 and PCA2 for *Erianthus procerus*-derived hybrids during the year 2018–2019. Chl: chlorophyll concentration; LA: leaf area; Flav: flavonoid index; NBI: nitrogen balance index; CY: cane yield; SCWT: single cane weight; G: germination; CD: cane diameter; B: Brix value; JE: juice extraction; PH: plant height; JW: juice weight; Pur: purity; NMCs: number of millable canes; Tiller 1: 60-day tillers; Tiller 2: 90-day tillers; Tiller 3: 120-day tillers.

**Figure 7 plants-13-01023-f007:**
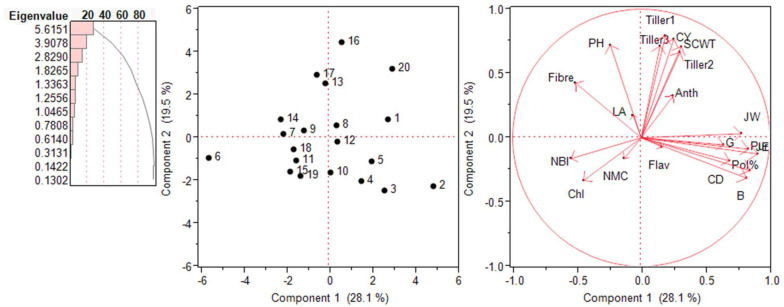
Scree plot, score plot, and loading plot of PCA1 and PCA2 for Erianthus procerus-derived hybrids during the year 2019–2020. Chl: chlorophyll concentration; LA: leaf area; Flav: flavonoid index; NBI: nitrogen balance index; CY: cane yield: SCWT: single cane weight; G: germination; CD: cane diameter; B: Brix % value in juice; JE: juice extraction; PH: plant height; JW: juice weight; Pur: purity; NMCs: number of millable canes; Tiller 1: 60-day tillers; Tiller 2: 90-day tillers; Tiller 3: 120-day tillers.

**Figure 8 plants-13-01023-f008:**
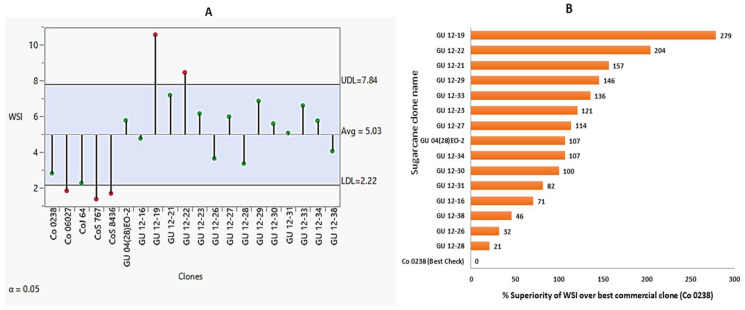
Winter sprout potential of *Erinathus procerus*-derived hybrids and commercial check varieties. (**A**): Mean of 3 years (2018–2019, 2019–2020, and 2020–2021) per the performance of the clones; (**B**): Mean of three years (2018–2019, 2019–2020, and 2020–2021) percent superiority of *Erinathus procerus*-derived hybrids over best commercial varieties.

**Figure 9 plants-13-01023-f009:**
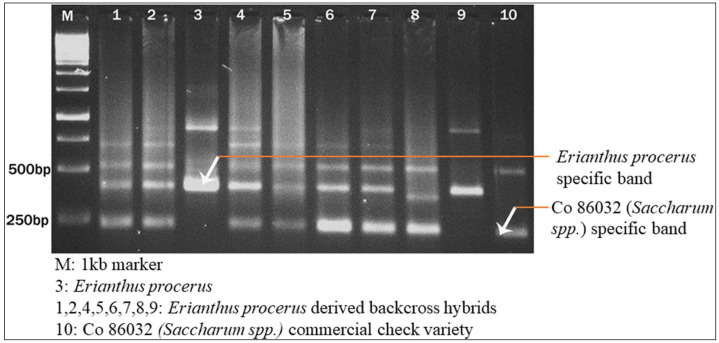
Confirmation of *Erianthus procerus*-derived hybrids using the *5S rDNA* marker.

**Table 1 plants-13-01023-t001:** Pooled analysis of variance for agro-morphological, quality, and physiological traits in *Erianthus procerus* clones and check varieties.

Traits	Main Effects				Interaction Effects	
	Genotypes (19)		Years (1)		Genotypes × Years (19)	
	MSS	*p* Value	MSS	*p* Value	MSS	*p* Value
Germination	216.4	0.0145 *	15,359.9	<0.0001 **	121.2	0.3211
60-day tillers	9.47 × 10^8^	<0.0001 **	2.61 × 10^9^	0.1845	3.18 × 10^8^	0.9997
90-day tillers	3.21 × 10^9^	<0.0001 **	1.11 × 10^11^	<0.0001 **	2.02 × 10^9^	0.0003 **
120-day tillers	3.30 × 10^9^	0.0005 **	2.16 × 10^11^	<0.0001 **	2.72 × 10^9^	0.0034 **
Plant height	9256.8	<0.0001 **	0.0	1.000	0.0	1.000
Cane diameter	0.7	<0.0001 **	0.0	1.000	0.0	1.000
Juice weight	2.5	<0.0001 **	0.0	1.000	6.95 × 102^9^	1.000
Juice extraction	397.7	<0.0001 **	0.0	1.000	4.21 × 102^9^	1.000
Brix value	15.1	<0.0001 **	6.8	0.0755	1.4	0.8373
Pol%	21.9	<0.0001 **	13.7	0.0272 *	2.5	0.5523
Purity	105.2	<0.0001 **	64.7	0.0969	24.5	0.4001
Single cane weight	0.2	<0.0001 **	0.01	0.6407	0.02	0.3455
Fibre content	49.9	<0.0001 **	45.6	0.0142 *	16.1	0.0076 **
Number of millable canes	1.42 × 10^9^	<0.0244 *	4.50 × 10^8^	0.4372	2.93 × 10^8^	0.9871
Cane yield	1495.1	<0.0001 **	689.8	0.1095	97.5	0.9913
Nitrogen balance index	213.7	<0.0001 **	70.9	0.1433	33.6	0.4334
Chlorophyll concentration	128.3	<0.0001 **	42.5	0.2199	26.5	0.5204
Flavonoid index	0.15	0.1072	0.2	0.1474	0.11	0.3385
Anthocyanin index	0.0008	0.2709	0.0004	0.4417	0.0006	0.5011
Leaf area	3020.1	0.748	1035.2	0.6119	1618.4	0.9852

The figure in parenthesis indicates the degree of freedom. ** Significance at *p* < 0.01; * Significance at *p* < 0.05.

**Table 2 plants-13-01023-t002:** Correlation matrix among cane growth parameters and physiological and quality traits.

	1	2	3	4	5	6	7	8	9	10	11	12	13	14	15	16	17	18	19	20
1	1.00	0.066	−0.45 **	−0.59 **	−0.01	0.12	0.17	0.21	0.17	0.06	0.05	−0.00	−0.37 *	−0.10	−0.08	−0.20	−0.17	0.23	−0.13	−0.18
2		1.00	0.57 **	0.40 *	0.66 **	−0.28	0.06	−0.09	−0.02	0.02	0.12	0.45 **	0.32 *	0.04	0.57 **	−0.04	−0.18	−0.24	0.02	−0.05
3			1.00	0.88 **	0.29	−0.15	0.00	−0.03	0.07	0.122	0.215	0.149	0.356 *	0.295	0.395 *	0.054	−0.090	−0.168	0.121	−0.007
4				1.00	0.207	−0.19	−0.09	−0.14	0.02	0.03	0.01	0.00	0.33 *	0.32 *	0.23	0.14	0.08	−0.21	0.11	−0.08
5					1.00	−0.14	0.13	−0.27	−0.31 *	−0.20	−0.09	0.43 **	0.43 **	−0.01	0.51 **	0.32 *	0.26	−0.25	−0.02	−0.22
6						1.00	0.85 **	0.81 **	0.50 **	0.51 **	0.47 **	0.28	−0.41 **	−0.18	0.20	−0.20	−0.17	0.09	0.12	0.16
7							1.00	0.86 **	0.66 **	0.67 **	0.58 **	0.50 **	−0.34 *	−0.27	−0.36 *	−0.19	−0.15	0.00	0.07	0.08
8								1.00	0.77 **	0.79 **	0.70 **	0.44 **	−0.50 **	−0.32 *	0.27	−0.40 *	−0.37 *	0.12	0.17	0.12
9									1.00	0.97 **	0.77 *	0.29	−0.57 **	−0.37 *	0.07	−0.35 *	−0.27	0.09	0.09	0.06
10										11.00	0.89 **	0.34 *	−0.54 **	−0.34 *	0.17	−0.35 *	−0.31	0.09	0.14	0.09
11											1.00	0.40 *	−0.36 *	−0.21	0.35 *	−0.29	−0.36 *	0.08	0.021	0.13
12												1.00	0.123	−0.53 **	0.84 **	−0.26	−0.29	−0.00	0.33 *	0.07
13													1.00	0.35 *	0.35 *	0.34 *	0.19	−0.23	−0.10	0.08
14														1.00	−0.01	0.26	0.04	0.00	−0.12	−0.11
15															1.00	−0.10	−0.27	−0.01	0.24	0.05
16																1.00	0.84 **	−0.47 **	−0.57 **	−0.01
17																	1.00	−0.19	−0.52 **	0.02
18																		1.00	0.31 *	−0.09
19																			1.00	−0.15
20																				1.00

1: Germination; 2: 60-day tillers; 3: 90-day tillers; 4: 120-day tillers; 5: Plant height; 6: Cane diameter; 7: Juice weight; 8: Juice extraction; 9: Brix value; 10: Pol%; 11: Purity; 12: Single cane weight; 13: Fibre content; 14: Number of millable canes; 15: Cane yield; 16: Nitrogen balance index; 17: Chlorophyll concentration; 18: Flavonoid index; 19: Anthocyanin index; 20: Leaf area. ** Significance at *p* < 0.01; * Significance at *p* < 0.05.

**Table 3 plants-13-01023-t003:** Winter sprouting index (WSI) of *Erianthus procerus*-derived hybrids and commercial checks during 2018–2019, 2019–2020, and 2020–2021.

SN	Genotype	Winter Sprouting Index (WSI)			
		2018–2019	2019–2020	2020–2021	Mean
1	GU 12—19	8.0	12.2	11.6	10.6
2	GU 12—22	6.1	9.9	9.5	8.5
3	GU 12—21	5.3	7.2	9.1	7.2
4	GU 12—29	3.5	8.8	8.4	6.9
5	GU 12—33	4.0	8.6	7.3	6.6
6	GU 12—23	4.8	7.3	6.4	6.2
7	GU 12—27	3.9	8.7	5.5	6.0
8	GU 12—34	4.2	6.9	6.3	5.8
9	GU 04(28)EO—2	6.0	6.0	5.3	5.8
10	GU 12—30	5.0	5.4	6.5	5.6
11	GU 12—31	3.1	5.3	6.9	5.1
12	GU 12—16	3.7	5.5	5.3	4.8
13	GU 12—38	2.5	4.1	5.7	4.1
14	GU 12—26	2.9	4.2	3.9	3.7
15	GU 12—28	2.9	4.0	3.3	3.4
16	Co 0238 (Check)	2.4	3.6	2.5	2.8
17	CoJ 64 (Check)	1.7	2.9	2.3	2.3
18	Co 06027 (Check)	0.5	3.1	2.0	1.9
19	CoS 8436 (Check)	1.0	2.5	1.7	1.7
20	CoS 767 (Check)	0.9	1.9	1.4	1.4

**Table 4 plants-13-01023-t004:** Reaction of *Erinathus procerus* derived clones for red rot against tropical and subtropical isolates during 2018–2019 and 2019–2020.

*Erinathus procerus*	2018–2019 and 2019–2020		2018–2019 and 2019–2020	
	Coimbatore (Tropical Isolates)		Karnal (Subtropical Isolates)	
Clone name	*Cf671*	*Cf671* + *Cf94012*	*Cf08*	*Cf09*
GU04 (28) EO-2	R	R	MR	MR
GU12—16	MR	R	S	MR
GU12—19	R	MR	MR	MR
GU12—21	R	R	MR	R
GU12—22	MR	MR	MR	MR
GU12—23	MR	R	-	MS
GU12—26	MR	MR	MR	MR
GU12—27	MR	R	MR	R
GU12—28	MR	MR	S	MR
GU12—29	MR	R	S	MR
GU12—30	MR	MR	MR	MR
GU12—31	MR	R	MR	MR
GU12—33	R	MR	MS	MR
GU12—34	R	MR	MS	S
GU12—38	S	HS	S	S
Co 06027	MR	S	S	MR

R: resistance; MR: moderately resistant; MS: moderately susceptible; S: susceptible.

**Table 5 plants-13-01023-t005:** Parentage of *Erianthus procerus* derived hybrids and check varieties used in the study.

SN	Clone	Generation	Female	Male	Remarks
1	GU 04(28) EO—2	F_1_	IND 90-776	PIO 96-435	GU 04 (28) EO—2 is an intergeneric hybrid between *Erianthus procerus* and *S. officinarum*
2	GU 12—16	BC_1_	GU 04(28) EO-2	Co 06027	BC_1_ hybrid
3	GU 12—19	BC_1_	GU 04(28) EO-2	Co 06027	BC_1_ hybrid
4	GU 12—21	BC_1_	GU 04(28) EO-2	Co 06027	BC_1_ hybrid
5	GU 12—22	BC_1_	GU 04(28) EO-2	Co 06027	BC_1_ hybrid
6	GU 12—23	BC_1_	GU 04(28) EO-2	Co 06027	BC_1_ hybrid
7	GU 12—26	BC_1_	GU 04(28) EO-2	Co 06027	BC_1_ hybrid
8	GU 12—27	BC_1_	GU 04(28) EO-2	Co 06027	BC_1_ hybrid
9	GU 12—28	BC_1_	GU 04(28) EO-2	Co 06027	BC_1_ hybrid
10	GU 12—29	BC_1_	GU 04(28) EO-2	Co 06027	BC_1_ hybrid
11	GU 12—30	BC_1_	GU 04(28) EO-2	Co 06027	BC_1_ hybrid
12	GU 12—31	BC_1_	GU 04(28) EO-2	Co 06027	BC_1_ hybrid
13	GU 12—33	BC_1_	GU 04(28) EO-2	Co 06027	BC_1_ hybrid
14	GU 12—34	BC_1_	GU 04(28) EO-2	Co 06027	BC_1_ hybrid
15	GU 12—38	BC_1_	GU 04(28) EO-2	Co 06027	BC_1_ hybrid
16	Co 06027	-	CoC 671	IG 91-1100	Commercial Variety Tropical
17	Co 0238	-	Co LK 8102	Co 775	Commercial Variety Subtropical
18	CoJ 64	-	Co 976	Co 617	Commercial Variety Subtropical
19	CoS 767	-	Co 419	Co 319	Commercial Variety Subtropical
20	CoS 8436	-	MS 68/47	Co 1148	Commercial Variety Subtropical

## Data Availability

All data supporting the conclusions of this manuscript are provided within the manuscript.

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
