# Peer review of "Deciphering Winter Sprouting Potential of Erianthus procerus Derived Sugarcane Hybrids under Subtropical Climates"

_plants, 2024, doi:10.3390/plants13071023_

Round 1

Reviewer 1 Report

Comments and Suggestions for Authors

Dear Authors,

The manuscript is well prepared. However, I recommend some changes in English. Also, to consider in the SPAD index represents the content of particular compounds. 

Comments on the Quality of English Language

I recommend the English editing due to style mistakes.

Author Response

As per the reviewer’s suggestion, we have gone through the manuscript and made the necessary changes in English. Also, the details of the physiological parameters measurement are given in the revised manuscript in section 4.4 of materials and methods part. Also included the corrections suggested by the reviewer in the attached file.

Reviewer 2 Report

Comments and Suggestions for Authors

Here are some points and areas that may need improvement in the report:

The structure and organization of the article could be improved for better readability. The information seems scattered, and it would benefit from a clearer flow and division into sections. Consider providing subheadings for different sections to guide readers through the content more effectively.

Some figures and tables lack detailed labels and captions, making it challenging to understand the information they are intended to convey. Figures 1, 2, 3, and 4 need detailed explanations in the main text. The reader should not have to guess the meaning of the presented graphs.

Ensure consistency in the terminology used throughout the article. For example, in some places, "Flavonoid" is used, while in others, "Flavanoid" is mentioned.

The significance values (p-values) presented in Table 2 lack context. Readers would benefit from a brief interpretation of the statistical significance and its implications for the study.

The discussion of results is somewhat limited. Consider expanding on the implications of the findings and their significance for the field. Why are certain traits significant, and how do they contribute to the overall understanding of Erianthus procerus hybrids?

The article contains several abbreviations and acronyms without proper explanation. Provide a glossary or explanations for these abbreviations to aid comprehension.

The discussion on winter sprouting ability (Section 2.5) lacks depth. Readers might need more information on the importance of winter sprouting ability, and the results could be discussed in the context of potential agricultural applications.

The explanation of the PCA results (Section 2.6) is minimal. Provide a more detailed interpretation of the loading plots and how they contribute to understanding the traits and genotypes.

The discussion on resistance against red rot (Section 2.9) could benefit from more context. Why is red rot resistance important, and how do the results contribute to the field?

The article lacks a clear conclusion summarizing key findings. Consider adding a conclusion section that highlights the main outcomes and potential future directions for research.

By addressing these points, the article can become more comprehensive and accessible to a broader audience.

Comments on the Quality of English Language

Moderate editing of English language required

Author Response

Response to reviewer comments

Reviewer # 2

The structure and organization of the article could be improved for better readability. The information seems scattered, and it would benefit from a clearer flow and division into sections. Consider providing subheadings for different sections to guide readers through the content more effectively.

Author’s Response: As per the reviewer’s suggestion, the structure and organization of the article have been improved by rearranging the information for better flow. Also, the content is now divided into sections and subsections.

Some figures and tables lack detailed labels and captions, making it challenging to understand the information they are intended to convey. The reader should not have to guess the meaning of the presented graphs.

Author’s Response: Agreed, the figures and tables are now provided with detailed labels and captions.

Figures 1, 2, 3, and 4 need detailed explanations in the main text.

Author’s Response: As per the reviewer’s suggestion, Figure 1 has been explained in detail in the main text in the revised manuscript in subsection 2.1 of the results part. Figures 2, 3, and 4 have been explained in detail in the main text in the revised manuscript in subsection 2.3.1 of the results part.

Ensure consistency in the terminology used throughout the article. For example, in some places, "Flavonoid" is used, while in others, "Flavanoid" is mentioned.

Author’s Response: Agreed, we have ensured the consistency of “Flavonoid” throughout the manuscript.

The significance values (p-values) presented in Table 2 lack context. Readers would benefit from a brief interpretation of the statistical significance and its implications for the study.

Author’s Response: As per the reviewer’s suggestion, details of statistical significance are given in section 2.2 for the result part.

The discussion of results is somewhat limited. Consider expanding on the implications of the findings and their significance for the field. Why are certain traits significant, and how do they contribute to the overall understanding of Erianthus procerus hybrids?

Author’s Response: As per the reviewer’s suggestion, the discussion part is elaborated.

The article contains several abbreviations and acronyms without proper explanation. Provide a glossary or explanations for these abbreviations to aid comprehension.

Author’s Response: As per the reviewer’s suggestion, we have provided a glossary or explanations for the abbreviations.

The discussion on winter sprouting ability (Section 2.5) lacks depth. Readers might need more information on the importance of winter sprouting ability, and the results could be discussed in the context of potential agricultural applications.

Author’s Response: As per the reviewer’s suggestion, the detailed discussion of winter sprouting ability along with its application for sugarcane breeding programs is given in section 3.2 of the discussion part.

The explanation of the PCA results (Section 2.6) is minimal. Provide a more detailed interpretation of the loading plots and how they contribute to understanding the traits and genotypes. 

Author’s Response: Agreed, a more detailed interpretation of PCA results is given in section 2.5 (section number changed after rearrangement of the manuscript).

The discussion on resistance against red rot (Section 2.9) could benefit from more context. Why is red rot resistance important, and how do the results contribute to the field?

Author’s Response: As per the reviewer’s suggestion, the importance of red rot resistance has been explained in detail in the main text in the revised manuscript in subsection 2.6.2 and also in the conclusion part.

The article lacks a clear conclusion summarizing key findings. Consider adding a conclusion section that highlights the main outcomes and potential future directions for research.

Author’s Response: As per the reviewer’s suggestion, the conclusion has been modified including key findings and future directions

Moderate editing of English language required

Author’s Response:  As per the reviewer’s suggestion, we have gone through the manuscript and made the necessary changes in English.

By addressing these points, the article can become more comprehensive and accessible to a broader audience.

Author’s Response: Agreed and we thank for valuable suggestions

Round 2

Reviewer 2 Report

Comments and Suggestions for Authors

Accept after minor revision (corrections to minor methodological errors and text editing)

Comments on the Quality of English Language

Minor editing of English language required

Author Response

Dear Reviewers

We thank you for improving our manuscript,  As per the reviewer’s suggestion, minor editing of the english language has been done, and also checked the manuscript thoroughly for typos and other minor errors. We hope that this revision may be considered for publication.

with kind regards

M R Meena

Sr Scientist
